# ANOMALY DETECTION IN DYNAMIC GRAPHS VIA ADVERSARIAL AUTOENCODER

## ABSTRACT

Anomaly detection in dynamic graphs is a very important task that has attracted a lot of attention. Many dynamic graph anomaly detection methods are already available, but most of these efforts are based on supervised learning. In the real world, however, it is often difficult to collect large amounts of labelled anomaly data, which is not conducive to the training of these supervised methods and severely reduces their ability to be applied in different dynamic graph anomaly detection scenarios. A novel semi-supervised anomaly detection framework **AAEDY** for the detection of anomalous edges in dynamic graphs is presented in this paper, which improves reconstruction by combining adversarial based on autoencoder, and discriminates whether an edge is anomalous by comparing the original edge to the reconstructed edge in low-dimensional space. Extensive experiments have been carried out on six real-world datasets, and the experimental results show that **AAEDY** can outperform the state-of-the-art competitors in anomaly detection significantly.

## 1 INTRODUCTION

In real life, dynamic graph networks can be seen everywhere, such as social networks, financial transaction networks, Internet of Vehicles, Internet of Things, etc. Among many dynamic graph analysis tasks, dynamic graph anomaly detection is one of the most important tasks, and its focus is on abnormal edge detection. Detecting abnormal edges helps to understand the state of the system, protect the system and improve its robustness (Ranshous et al., 2015; Wang et al., 2022). For example, in financial transaction systems, there are often criminals who try to make profits through fraud, phishing, money laundering, etc., which seriously endangers the interests of ordinary users and the healthy development of the system. Detecting abnormal edges in dynamic graphs is crucial to protecting the interests of ordinary users and maintaining the healthy development of the system.

At present, most of the graph anomaly detection methods based on deep learning are aimed at anomaly detection in static graphs, which are mainly divided into three methods: supervised (Chouiekh & Haj, 2018; Alsheikh et al., 2016; Perozzi et al., 2014; Grover & Leskovec, 2016; Tang et al., 2015), semi-supervised (Kumagai et al., 2021; Zhang et al., 2017; Castellini et al., 2017; Guo et al., 2016) and unsupervised (Li et al., 2017; Ding et al., 2019; Fan et al., 2020; Ding et al., 2021). Most of the supervised methods are based on the idea of feature embedding, which uses deep learning to embed nodes from high-dimensional space to low-dimensional space to obtain more effective node features. Semi-supervised and unsupervised methods are mostly based on the idea of residual analysis, which finds abnormal target edges or nodes by comparing the errors between the original data and the reconstructed data. Compared with anomaly detection methods on static graphs, there are fewer anomaly detection methods based on deep learning for dynamic graphs, and most of them are supervised methods. NetWalk Yu et al. (2018) uses random walks and deep autoencoders to obtain low-dimensional embeddings of nodes and combines clustering technology to detect anomaly information. StrGNN (Cai et al., 2021), H-VGRAE (Yang et al., 2020) and TADDY (Liu et al., 2021) design end-to-end supervised methods for anomaly detection. AddGraph (Zheng et al., 2019) designs a semi-supervised learning framework based on residual analysis.

However, dynamic graph anomaly detection based on supervised learning methods requires a large amount of labeled anomaly data to train the model. In practical applications, it is difficult to collect a large amount of anomaly data, which seriously reduces the application ability in different dy-

namic graph anomaly detection scenarios. In order to solve the problem that current dynamic graph anomaly detection methods are not very practical, **AAEDY**, a novel semi-supervised adversarial proposes and dynamic graph anomaly detection framework based on autoencoder, which only uses normal data to train the model. Specifically, the subgraph sampling method (Liu et al., 2021) is used to obtain the subgraph under the time snapshot of the target edge, and the node set is obtained according to the importance of nodes under multiple subgraphs to represent the target edge. Three encoding methods are used to encode the nodes from the global, local and temporal dimensions, and anomaly detection is performed by comparing the differences between the original edge and the reconstructed edge. In summary, the main contributions of this paper are as follows.

- A novel semi-supervised anomaly detection framework for dynamic graphs is proposed, which requires only normal data to train the model, solving the problem of difficulty in collecting anomaly information in real-world dynamic networks and effectively improving the applicability in different dynamic graph anomaly detection scenarios.

- An adversarial autoencoder based on residual analysis is designed, which effectively improves the reconstruction effect of the original edges by combining adversarial and autoencoder.

- Experiments on six real-world datasets achieve state-of-the-art performance, which proves the effectiveness of **AAEDY** on detecting anomalies in different kinds of graphs.

## 2 RELATED WORK

### 2.1 ANOMALY DETECTION IN STATIC GRAPHS

The structure, attributes and other features of the graph in the static graph do not change over time. A considerable number of methods have been developed for anomaly detection in static graphs using deep learning, which are mainly divided into supervised methods, semi-supervised methods and unsupervised methods. Among the supervised methods, Chouiekh & Haj (2018) proposed a supervised learning method that uses convolutional neural networks to extract user attribute features to distinguish normal users from abnormal users in telephone networks. Alsheikh et al. (2016) proposed a method for extracting features using deep learning on Spark computing nodes, which solves the problem that deep learning models contain a large number of hidden layers and a large number of parameters. However, supervised methods require a certain amount of labeled data to train the model, which is not suitable for real-world scenarios where anomalies are scarce.

Most semi-supervised and unsupervised methods adopt the idea based on residual analysis and use reconstruction errors to measure whether a data instance is abnormal. Li *et al.* proposed an anomaly detection framework Radar in the literature (Li et al., 2017) to detect anomalies in attribution networks from the perspective of residual analysis. Ding et al. (2019) designed a novel autoencoder framework and used GCN (Kipf & Welling, 2017) to capture the nonlinearity of data and the complex interactions between nodes. Fan et al. (2020) proposed a dual autoencoder framework, using structural autoencoders and attribute autoencoders to learn the structural and attribute features of static graphs. Ding et al. (2021) used generative adversarial methods to enhance the reconstruction ability of the model, and through joint training, the model automatically generated potential abnormal data, effectively solving the problem of inductive learning. Kumagai et al. (2021) proposed a semi-supervised static graph anomaly detection method that comprehensively considers the graph structure and node attribute information, and identifies abnormal nodes by learning hyperspheres. Zhang et al. (2017) proposed an autoencoder-based anomaly detection method for rumor detection. The focus of this method is to propose several adaptive thresholds to improve rumor detection. Castellini et al. (2017) proposed a semi-supervised deep learning model for anomaly detection, mainly using an autoencoder with a denoising mechanism to detect robot followers in social networks.

### 2.2 ANOMALY DETECTION IN DYNAMIC GRAPHS

The structure, attributes, and other features of dynamic graphs change over time, and it is necessary to comprehensively consider the graph information under multiple time snapshots. Miz et al. (2019) proposed a scalable community-based anomaly detection method that connects and aggre-

gates nodes with similar behaviors to analyze abnormal behaviors that occur in collective behaviors. NetWalk (Yu et al., 2018) encodes the vertices of a dynamic network into vector representations through Clique embedding, jointly minimizes the pairwise distance of each vertex representation, and combines clustering-based methods with low-dimensional vertex features to gradually dynamically detect network anomaly detection. Zheng et al. (2019) proposed a semi-supervised anomaly detection method for dynamic graphs, using extended time GCN to obtain the structural embedding and attribute embedding of the graph, and combining attention-based GRU (Chung et al., 2014) and negative sampling strategies to solve the problem of insufficient labels. Cai et al. (2021) extracts subgraph features under time snapshots through GCN, and uses recurrent neural network GRU to fuse multiple subgraph features to capture the temporal relationship of dynamic graphs. Liu et al. (2021) uses graph diffusion technology to sample substructures in dynamic graphs, adopts three encoding methods to capture the global structural features, local structural features and temporal information of the graph, and fuses multiple features through autoencoders to achieve anomaly detection of dynamic graphs. In order to cope with the problem of scarce abnormal data, **AAEDY** proposed a novel semi-supervised adversarial autoencoder, which only needs to use normal data in the model training stage, avoiding the problem of being unable to train the model due to lack of abnormal data.

## 3 FRAMEWORK

A novel and effective dynamic graph anomaly detection framework is proposed. The overall structure is shown in Figure 1. First, the graph diffusion technique is used to perform sub-graph sampling on the target edge to obtain the graph information of the target edge at different times, and three encoding methods are used to encode the dynamic graph information at multiple times. Secondly, a new deep autoencoder based on residual analysis is designed to detect abnormal edge information by comparing the errors between the original data and the reconstructed individual data, and the adversarial idea is used to continuously optimize the reconstruction performance training of the deep autoencoder.

### 3.1 DATA PREPROCESSING

In sampling, each edge in the dynamic graph is first sampled according to its centre, which is the centre of each subgraph after sampling, and is denoted as the target edge. The two nodes directly connected to the target edge are called target nodes, while the neighbouring nodes around the target node we call contextual nodes. The specific sampling steps are divided into static graph sampling under a single temporal snapshot and dynamic graph sampling under multiple temporal snapshots.

Static graph sampling under a single temporal snapshot uses graph diffusion techniques (Hassani & Khasahmadi, 2020; Klicpera et al., 2019) to collect a certain number of importance-aware contextual node sets against a target edge. The specific idea is to obtain the diffusion degree of each node for a target edge in a global view by using the graph diffusion technique, and use the diffusion degree as the node importance, and then sample the nodes according to their importance degree.

Formally, given an adjacency matrix of a static graph $\mathcal{G} \in \mathbb{R}^{n \times n}$, the graph diffusion $\mathcal{Q} \in \mathbb{R}^{n \times n}$ is defined as:

$$\mathcal{Q} = \sum_{k=0}^{\infty} \theta_k T^k \tag{1}$$

where $T \in \mathbb{R}^{n \times n}$ is the generalized transition matrix and $\theta_k$ is the weighting coefficient which determines the ratio of global-local information.

Graph diffusion matrix $\mathcal{Q}$ is built using the graph diffusion technique for a static graph with a single timestamp. For a given node, the connectivity with other nodes can be easily obtained through the diffusion matrix, and a connectivity vector can be composed from this. For a given target edge $e_{\text{tgt}} = (v_1, v_2)$, this can be obtained by summing the two target node vectors . The formula is as follows:

$$C_{e_{\text{tgt}}} = C_{v_1} + C_{v_2} \tag{2}$$

Then, the set of connectivity top-k nodes is then selected from this as the set of contextual nodes $\mathcal{U}(e_{\text{tgt}})$.

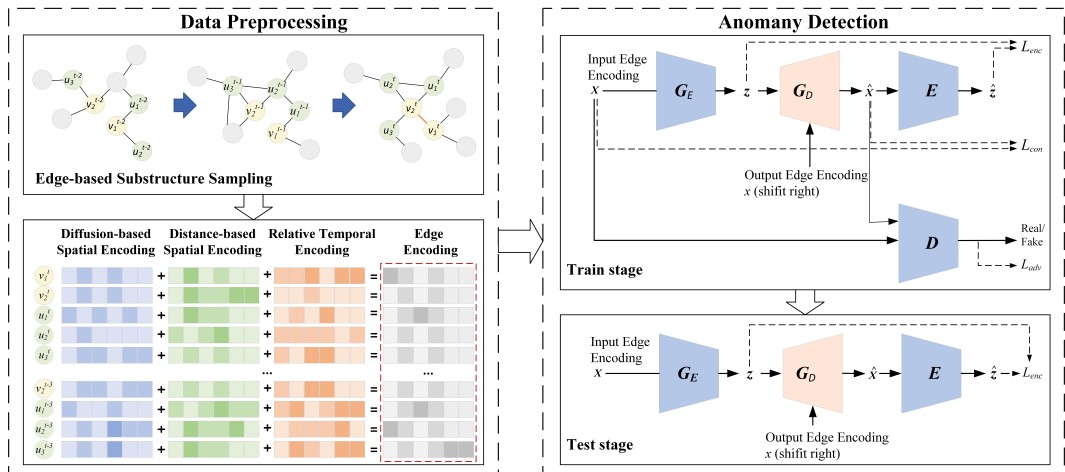

Figure 1: The architecture of our proposed **AAEDY** framework.

When it comes to dynamic graphs, using a time-sliding window mechanism, for a given target edge, obtain a sequence of graph information at different timestamps, and for each timestamp, use graph diffusion techniques to construct a diffusion matrix, then obtain the top-k important nodes by a static graph method at a single timestamp, and finally fuse the nodes at different timestamps into a total set of nodes $\mathcal{S}\left(e_{\text{tgt}}^t\right) = \bigcup_{i=t-\tau+1}^{t} \mathcal{S}^i\left(e_{\text{tgt}}^t\right)$.

### 3.1.1 DIFFUSION-BASED SPATIAL ENCODING

Diffusion-based spatio encoding is a method that uses the diffusion information of nodes at the global level for a secondary representation of each node. This method allows the use of global information to represent the target edges.

Specifically, since the target edge is represented by its node set $v_j^i \in \mathcal{S}^i\left(e_{\text{tgt}}^t\right)$ in each time-stamped subgraph, the diffusion value of each node in the node set can first be obtained by using a graph diffusion technique, then the diffusion value can be divided into ranks by a rank function, and finally these ranks can be mapped to a high-dimensional space using a learnable encoding function using a location encoding method very similar to that in natural language processing. The formula is as follows:

$$F_{\text{diff}}\left(v_j^i\right) = \text{linear}\left(\text{rank}\left(C_{e_{\text{tgt}}}^i\left[idx\left(v_j^i\right)\right]\right)\right) \in \mathbb{R}^{d_{enc}} \tag{3}$$

where $d_{enc}$ denotes the dimension of node encoding, linear($\cdot$) is the learnable encoding function, rank($\cdot$) denotes the rank calculation function, and idx($\cdot$) function is the index enquiring function.

### 3.1.2 DISTANCE-BASED SPATIAL ENCODING

Distance-based spatio encoding is similar to the idea of word position coding in natural language processing, by which the local structural information of a node can be captured so that each node has its unique representation in the set of nodes $v_j^i \in \mathcal{S}^i\left(e_{\text{tgt}}^t\right)$.

In practice: the distances of the local context nodes can be calculated by computing their distances to the two target nodes and taking the smaller of them. Their distances to the two target nodes and taking the smaller of them. And the distance from the target node to the target edge is set to 0. Finally these local importance are then mapped to a higher dimensional space by an encoding function that can be learned. The formula is as follows.

$$F_{\text{dist}}\left(v_j^i\right) = \text{linear}\left(\min\left(\text{dist}\left(v_j^i, v_1^i\right), \text{dist}\left(v_j^i, v_2^i\right)\right)\right) \in \mathbb{R}^{d_{\text{enc}}} \tag{4}$$

where dist($\cdot$) function is the relative distance calculation function, min($\cdot$) function serves to calculate the minimum value. The linear($\cdot$) and $d_{enc}$ have the same meaning as Equation 3.

### 3.1.3 RELATIVE TEMPORAL ENCODING

Since nodes may come from different single temporal snapshot in the set of nodes $v_j^i \in \mathcal{S}^i\left(e_{\text{tgt}}^t\right)$ representing the target edge, encoding using the relevant time sets allows the representation of each node's time information, thus allowing the model to capture the important factor of dynamic variability of the target edge.

Represent the temporal information of the node by computing the difference between the time $t$ at which the target edge is located and the time $i$ at which the node is located, and map these temporal differences to a high-dimensional space using the learnable encoding function mentioned in the previous section. The formula is as follows:

$$F_{\text{temp}}\left(v_j^i\right) = \text{linear}(\|t - i\|) \in \mathbb{R}^{d_{\text{enc}}} \tag{5}$$

where $\|\cdot\|$ is the relative time computing function, $\text{linear}(\cdot)$ and $d_{enc}$ have the same meaning as Equation 3.

### 3.1.4 EDGE ENCODING

Once the three terms are computed, they are merged to form the downstream adversarial autoencoder model's input edge encodings. For performance reasons, the three encoding terms are added together rather than concatenate them into a larger vetor. The encoding fusion is formalized as follows:

$$F\left(v_j^i\right) = F_{\text{diff}}\left(v_j^i\right) + F_{\text{dist}}\left(v_j^i\right) + F_{\text{temp}}\left(v_j^i\right) \in \mathbb{R}^{d_{\text{enc}}} \tag{6}$$

Finally, for the target edge $e_{\text{tgt}}^t$, we stack all the node encodings in the node set into an encoding matrix, which represents the feature information of the target edge and serves as the input information for the anomaly detection model proposed in the next section. The encoding matrix is represented by:

$$X\left(e_{\text{tgt}}^t\right) = \bigoplus_{v_j^i \in \mathcal{S}\left(e_{\text{tgt}}^t\right)} \left[F\left(v_j^i\right)\right]^\top \in \mathbb{R}^{(\tau(k+2)) \times d_{enc}} \tag{7}$$

where $[\cdot]^\top$ is the transpose operation and $\bigoplus$ is the concatenation operation.

## 3.2 ANOMANY DETECTION

Anomalous edge detection in dynamic graphs by designing an adversarial autoencoder. This autoencoder consists of a generator $G$, an encoder $E$, and a discriminator $D$.

The generator $G$ is a autoencoder, consisting of an encoder $G_E$, and a decoder $G_D$, which is mainly based on transformer's self-attention mechanism and multi-headed attention mechanism (Vaswani et al., 2017) to learn how to reconstruct the original edges of the input. Unlike Transformer, we add an embedding layer to embed the high-dimensional features of the original edges into the low-dimensional space. This generator $G$ works as follows: the generator $G$ first reads an input edge $x$ before forwarding it to its encoder network $G_E$. By adding an embedding layer to the transformer, $G_E$ embeds the whole input edge $x$ into a low-dimensional vector $z$, which is also referred to as the bottleneck features of $G$, since it has the smallest dimension that contains the best representation of $x$. The decoder part $G_D$ of the generator network $G$ adopts the architecture of transformer's decoder, on this basis, a layer of neural network is added to restore the low-dimensional vector $z$ to its size before embedding. This approach upscales the vector $z$ to reconstruct the edge $x$ as $\hat{x}$.

The encoder $E$ can take the $\hat{x}$, generated by the generator $G$ and re-embed it and generate the embedding vector $\hat{z}$. The network structure of the encoder is the same as that of the encoder $G_E$ in the generator, so that the generated embedding vector $\hat{z}$, has the same spatial dimension as the generator embedding vector $z$.

The discriminator $D$ whose objective is to classify the input $x$ and the output $\hat{x}$ as real or fake, respectively. In our discriminator network, a layer of self-attention network is used to capture the features of the edge $x$ or $\hat{x}$, and add a layer of linear neural network to map them to a one-dimensional space. It is also normalized and nonlinearized by Softmax function as the activation function.

### 3.2.1 Model Training

In the training phase of the model, trained with a large amount of normal data, the generator $G_D$ can reconstruct the normal data well and then pass the encoder $E$. Then the embedding vector $z$ and the embedding vector $\hat{z}$ have a small difference.

It is hypothesized that when an abnormal edge is forward-passed into the network $G$, $G_D$ does not be able to reconstruct the abnormalities although $G_E$ can map the input $x$ to the latent vector $z$. This should be expected because the network is modeled exclusively by normal samples during training and its parametrization makes the network unsuitable for generating abnormal samples. There is a potential consequence of an output $\hat{x}$ that has missed abnormalities in it, resulting in the encoder's network $E$ mapping mapping $\hat{x}$ to a vector $\hat{z}$ with the same lack of representation of the abnormal features,resulting in a dissimilarity between $z$ and $\hat{z}$ causing dissimilarity between $z$ and $\hat{z}$. As result, if there is some dissimilarity within latent vector space corresponding to an input edge $x$, the model classifies $x$ as an anomalous edge. To validate this hypothesis, the objective function was formulated by combining three loss functions, each optimised for a single sub-network.

**Adversarial Loss**. As a result of back propagation, the generator parameters are updated under the supervision of the cross entropy loss function. The discriminatory ability of the discriminator is calculated by the cross-entropy loss function, and the smaller the loss value, the stronger the discriminatory ability of the discriminator.

Hence, the adversarial loss $\mathcal{L}_{adv}$ is defined as:

$$\mathcal{L}_{adv} = \mathbb{E}_{x \sim p_X} \|D(x) - D(G(x))\|_{cross}. \tag{8}$$

**Contextual Loss**. It is sufficient to fool the discriminator $D$ with generated samples by using the adversarial loss $L_{adv}$. Even so, with only an adversarial loss, the generator would not be optimized towards learning contextual information about the input data. In order to fix this, the distance between the input and generated edges is measured to penalize the generator. Here the difference between the input and generated edges is calculated using a non-log-likelihood function. The contextual loss $\mathcal{L}_{con}$ is defined as:

$$\mathcal{L}_{con} = \mathbb{E}_{x \sim p_X} \|x - G(x)\|_{-log} \tag{9}$$

---

**Algorithm 1** Model Train Algorithm

---

**Input:** Training set of dynamic graph: $\mathbb{G}_{train} = \{\mathcal{G}^t\}_{t=1}^T$, Number of training epoches: $n$, Number of sampled contextual nodes: $k$, Size of time window: $\tau$, Number of attention heads $h$, Number of attention Layers $r$.
1: Randomly initialize the parameters of encoding linear mappings, Adversarial autoencoder model.
2: **for** $i \in 1, 2, \cdots, n$ **do**
3:     Split the dynamic graph $\mathbb{G}_{train} = \{\mathcal{G}^t\}_{t=1}^T$ with maximum timestamp $T$ into snapshots at $\tau$ timestamps $\mathcal{G}_{train}^t = (\mathcal{V}^t, \mathcal{E}^t) \in \{\mathcal{G}^t\}_{t=\tau}^T$
4:     **for** $e \in \mathcal{E}^t$ **do**
5:         Set $e$ the as the target edge and sample its subgraph node set $\mathcal{S}(e)$ with $\tau(k+2)$ nodes
6:         Calculate edge encoding matrix $X(e)$ via Equation (3)-(7)
7:         **for** m = 1 to 10 (or other numbers) **do**
8:             Calculate the total loss value $\mathcal{L}$ via Equation (8)-(11)
9:             Back propagation and update the generator parameters
10:         **end for**
11:         Calculate the loss value $\mathcal{L}_{adv}$ of the discriminator $D$ via Equation (8)
12:         Back propagation and update the discriminator parameters
13:     **end for**
14: **end for**

---

**Encoder Loss**. Using the two losses described above, the generator can be forced to produce a set of edges that are not only realistic, but also contextually appropriate. Further, a second encoder loss function $\mathcal{L}_{enc}$ is used, which uses a $\mathcal{L}_2$ loss function to minimize the distance between the bottleneck

features of the input $z = G_E(x)$ and the encoded features of the generated edge $\hat{z} = E(G(x))$. The $\mathcal{L}_{enc}$ is formally defined as:

$$\mathcal{L}_{enc} = \mathbb{E}_{x \sim p_X} \|G_E(x) - E(G(x))\|_2 \tag{10}$$

In so doing, the generator learns how to encode features of the generated image for normal samples. To improve the robustness of the model, the model is trained using these three loss functions and the three loss functions are added together to form a single total loss function. Overall, the objective function for the generator becomes the following:

$$\mathcal{L} = \mathcal{L}_{adv} + \mathcal{L}_{con} + \mathcal{L}_{enc} \tag{11}$$

Algorithm 1 gives the overall training process of the model.

### 3.2.2 MODEL TESTING

In the testing phase, when anomalous data is required, the generator cannot effectively reconstruct $x$. The $\hat{x}$ and $x$ have a certain discrepancy, which can be further amplified by embedding it through the encoder $E$, because the model has not seen this type of data before.

During the test stage, the model employs the $\mathcal{L}_{enc}$ function from equation (10) to determine the abnormality of a given edge. As a result, for a test sample $x_{test}$, our anomaly score $\mathcal{A}(x_{test})$ is defined as:

$$\mathcal{A}(x_{test}) = \|G_E(x_{test}) - E(G(x_{test}))\|_2 \tag{12}$$

## 4 EXPERIMENTS

### 4.1 EXPERIMENT SETTING

The **AAEDY** framework is tested on six real-world dynamic graph benchmark datasets, including UCI Messagegs (Opsahl & Panzarasa, 2009), Bitcoin-Alpha (Kumar et al., 2016), Bitcoin-OTC (Kumar et al., 2018), Digg (De Choudhury et al., 2009), AS-Topology (Zhang et al., 2005), and Email-DNC (Rossi & Ahmed, 2015).We compared **AAEDY** with seven anomaly detection methods, including DeepWalk (Perozzi et al., 2014) , node2vec (Grover & Leskovec, 2016) , spectral clustering (Opsahl & Panzarasa, 2009) , NetWalk (Yu et al., 2018) , AddGraph (Zheng et al., 2019) , StrGNN (Cai et al., 2021) , and TADDY (Liu et al., 2021).

Model training requires a large amount of normal data. Therefore, in the experiment, the training set and test set are divided into 80% and 20%. The training set contains all normal data, while the test set is injected with 1%, 5%, and 10% of abnormal data through the anomaly injection method. The test indicator uses AUC, which is insensitive to the balance of positive and negative samples and can be reasonably evaluated in the case of sample imbalance. In the preprocessing stage, the number of context nodes is set to 5, the time window $\tau$ is set to 2, and the encoding dimension $d_{enc}$ is set to 512. The PPR (Page et al., 1999) diffusion is the same as that used in TADDY and is calculated by Equation (2). In the anomaly detection stage, in all datasets, the number of attention heads is set to 1, the number of attention layers is set to 1, and the dimensions of $z$ and $\hat{z}$ are set to 10. The model is trained by the Adam optimizer with a learning rate of 0.0001. On the UCI Messages, Digg, and Email-DNC datasets, 0.01 is set as the anomaly threshold.

In order to balance the performance between the generator $G$ and the discriminator $D$, during the training process, the generator $G$ is trained first, and then the discriminator $D$ is trained, and the weights between the generator, encoder, and discriminator can be dynamically adjusted. Different from the above datasets, a new training strategy is used on Bitcoin-OTC, Bitcoin-Alpha, and AS-Topology datasets to train the generator $G$ and encoder $E$ once for each discriminator $D$. One epoch. 20 epochs are trained on all datasets, and the best trained model is finally adopted. The snapshot size is set to 1,000 for UCI Messages and Email-DNC, 2,000 for Bitcoin-Alpha and Bitcoin-OTC, and 4,000 for AS-Topology and Digg, respectively.

### 4.2 RESULTS ON BENCHMARK DATASETS

Experiments are carried out on six real data sets separately. The experimental results are shown in Table 1. It can be found that our method has achieved better performance compared with the best

Table 1: AUC comparison on benchmark datasets.

| Methods | UCI Messages | | | Bitcoin-Alpha | | | Bitcoin-OTC | | |
|---|---|---|---|---|---|---|---|---|---|
| | 1% | 5% | 10% | 1% | 5% | 10% | 1% | 5% | 10% |
| DeepWalk | 0.7514 | 0.7391 | 0.6979 | 0.6958 | 0.6874 | 0.6793 | 0.7423 | 0.7356 | 0.7287 |
| node2vec | 0.7371 | 0.7433 | 0.6960 | 0.6910 | 0.6802 | 0.6785 | 0.6951 | 0.6883 | 0.6745 |
| Spectral Clustering | 0.6324 | 0.6104 | 0.5794 | 0.7401 | 0.7275 | 0.7167 | 0.7624 | 0.7376 | 0.7047 |
| NetWalk | 0.7758 | 0.7647 | 0.7226 | 0.8385 | 0.8357 | 0.8350 | 0.7785 | 0.7694 | 0.7534 |
| AddGraph | 0.8083 | 0.8090 | 0.7688 | 0.8665 | 0.8403 | 0.8498 | 0.8352 | 0.8455 | 0.8592 |
| StrGNN | 0.8179 | 0.8252 | 0.7959 | 0.8574 | 0.8667 | 0.8627 | 0.9012 | 0.8775 | 0.8836 |
| TADDY | 0.8912 | 0.8398 | 0.8370 | 0.9451 | 0.9341 | 0.9423 | 0.9455 | 0.9340 | 0.9425 |
| **AAEDY** | **0.9521** | **0.9513** | **0.9524** | **0.9683** | **0.9735** | **0.9758** | **0.9781** | **0.9642** | **0.9756** |

| Methods | Digg | | | Email-DNC | | | AS-Topology | | |
|---|---|---|---|---|---|---|---|---|---|
| | 1% | 5% | 10% | 1% | 5% | 10% | 1% | 5% | 10% |
| DeepWalk | 0.7080 | 0.6881 | 0.6396 | 0.7481 | 0.7303 | 0.7197 | 0.6844 | 0.6793 | 0.6682 |
| node2vec | 0.7364 | 0.7081 | 0.6508 | 0.7391 | 0.7284 | 0.7103 | 0.6821 | 0.6752 | 0.6668 |
| Spectral Clustering | 0.5949 | 0.5823 | 0.5591 | 0.8096 | 0.7857 | 0.7759 | 0.6685 | 0.6563 | 0.6498 |
| NetWalk | 0.7563 | 0.7176 | 0.6837 | 0.8105 | 0.8371 | 0.8305 | 0.8018 | 0.8066 | 0.8058 |
| AddGraph | 0.8341 | 0.8470 | 0.8369 | 0.8393 | 0.8627 | 0.8773 | 0.8080 | 0.8004 | 0.7926 |
| StrGNN | 0.8162 | 0.8254 | 0.8272 | 0.8775 | 0.9103 | 0.9080 | 0.8553 | 0.8352 | 0.8271 |
| TADDY | 0.8617 | 0.8545 | 0.8440 | 0.9348 | 0.9257 | 0.9210 | 0.8953 | 0.8952 | 0.8934 |
| **AAEDY** | **0.8873** | **0.8935** | **0.8942** | **0.9411** | **0.9565** | **0.9458** | **0.9081** | **0.9142** | **0.9151** |

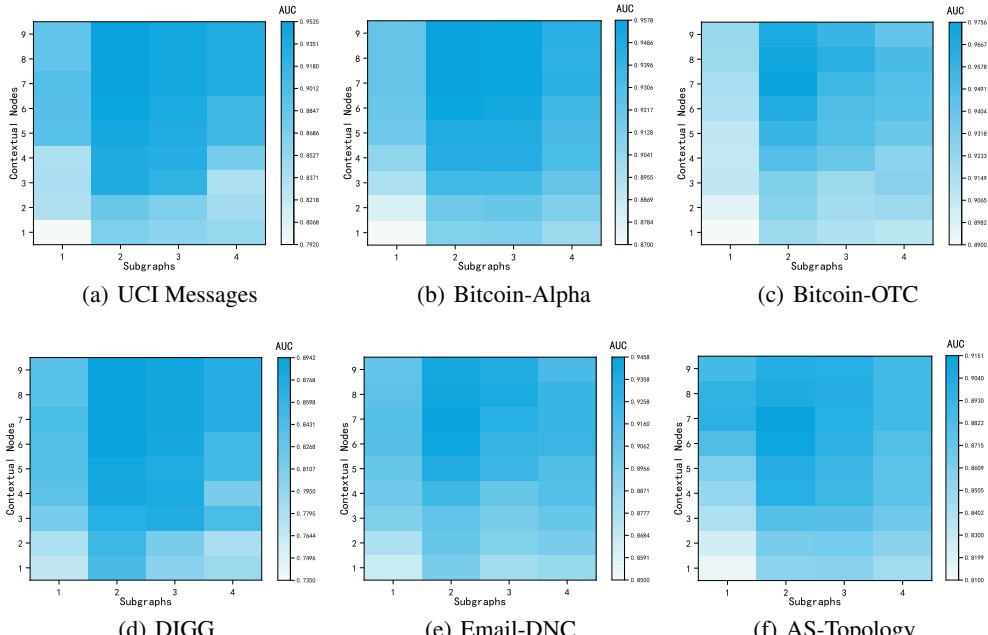

(a) UCI Messages  (b) Bitcoin-Alpha  (c) Bitcoin-OTC

(d) DIGG  (e) Email-DNC  (f) AS-Topology

Figure 2: The sensitivity of the number of subgraphs and the number of contextual nodes on six datasets.

baseline when injecting 1%, 5% or 10% abnormalities. Experimental results show that our method can effectively detect anomalies in dynamic graphs.

### 4.3 PARAMETER SENSITIVITY

This section analyses the effect of hyperparameters on the performance of **AAEDY**, focusing on the effect of the number of attention and attention layers and the training ratio on performance.

Firstly, on the UCI Messages dataset, Bitcoin-Alpha dataset, Bitcoin-OTC dataset, DIGG dataset, Email-DNC dataset and AS-Topology dataset, the number of subgraphs and the number of contextual nodes are set as $\{1, 2, 3, 4\}$ and $\{1, 2, 3, 4, 5, 6, 7, 8, 9\}$ respectively, and conducted a large number of experiments. In the experiment, the training set is set to account for 80% , the number of epochs is 20, and the two different training strategies mentioned above are used. The experimental

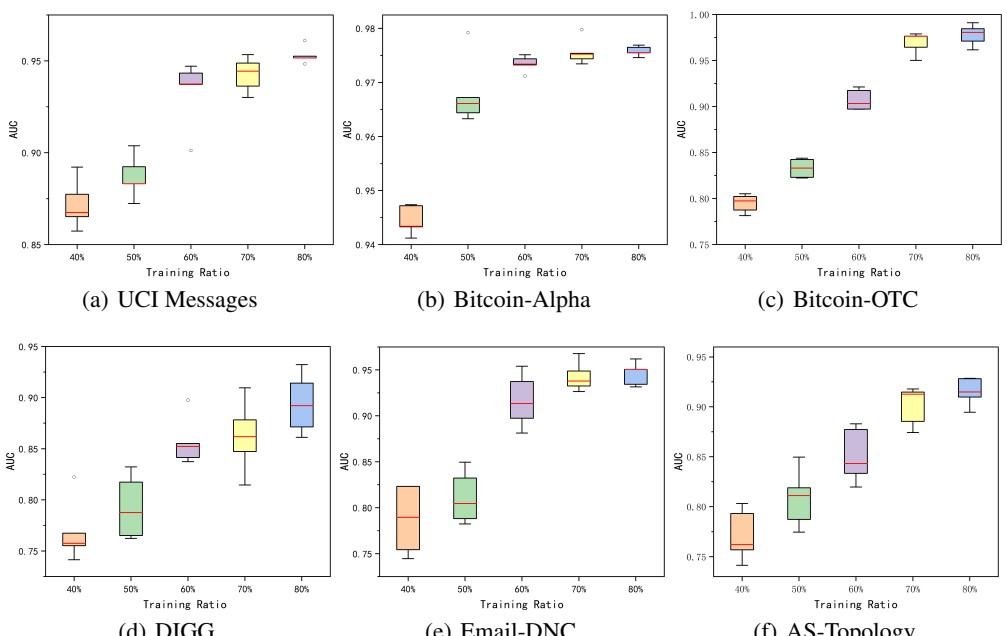

Figure 3: The sensitivity of training ratios on six datasets.

results are shown in Figure 2. It can be seen that it gives the best results with a number of subgraphs of 2 and 3 and a number of contextual nodes of around 7.

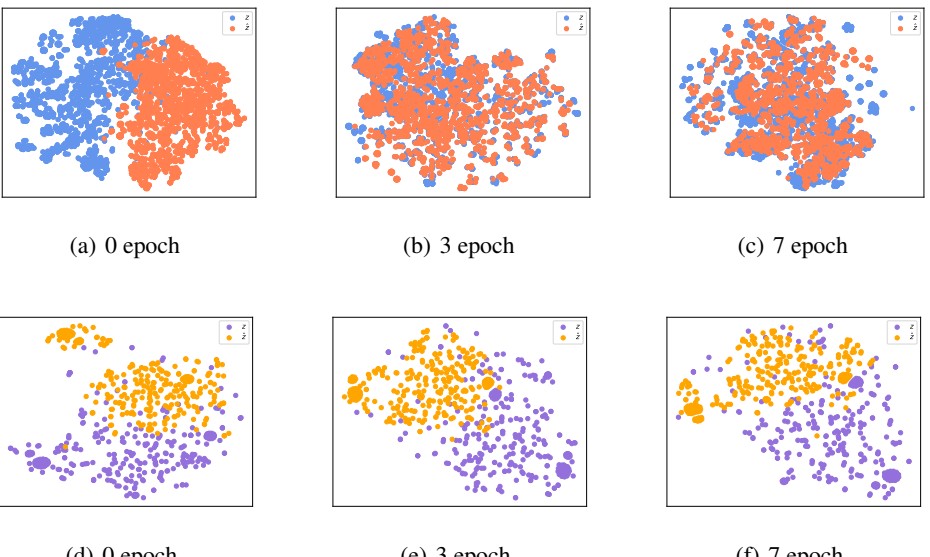

Figure 4: Visualisation of reconstruction differences between normal and abnormal edges on the UCI dataset. The first row shows the normal edge visualisation, the second row shows the abnormal edge visualisation.

Then, the effect of training ratio on performance is analyzed on the UCI Messages dataset, Bitcoin-Alpha dataset, Bitcoin-OTC dataset, DIGG dataset, Email-DNC dataset and AS-Topology dataset where the ratio of the training is set to $\{40\%, 50\%, 60\%, 70\%, 80\%\}$ and inject 10% of anomalous data into the test set. In the experiments, the number of epochs is set to 10 , using the two different training strategies mentioned above. The experimental results are shown in Figure 3, and it can be seen when the effect of anomaly detection is proportional to the training ratio.

## 4.4 VISUAL ANALYTICS

In this section, the embedding vectors $z$, $\hat{z}$ of normal and abnormal edges are visualised separately. And the individual vectors are downscaled to a two-dimensional space using TSNE to visualise the degree of difference between edges $z$ and $\hat{z}$ by visualising the spatial distance. Experiments are conducted on the UCI dataset, and the visualisation results of 0 epoch, 3 epoch and 7 epoch are shown in Figure 4. It can be found that for normal edges, the embedding vectors $z$ and $\hat{z}$ are relatively close in spatial distribution, while for abnormal edges the feature vectors $z$ and $\hat{z}$ are farther apart in spatial distribution, therefore, this aspect can be used to identify abnormal edges by reconstructing the differences.

## 4.5 MULTI-INDICATOR COMPARATIVE ANALYSIS

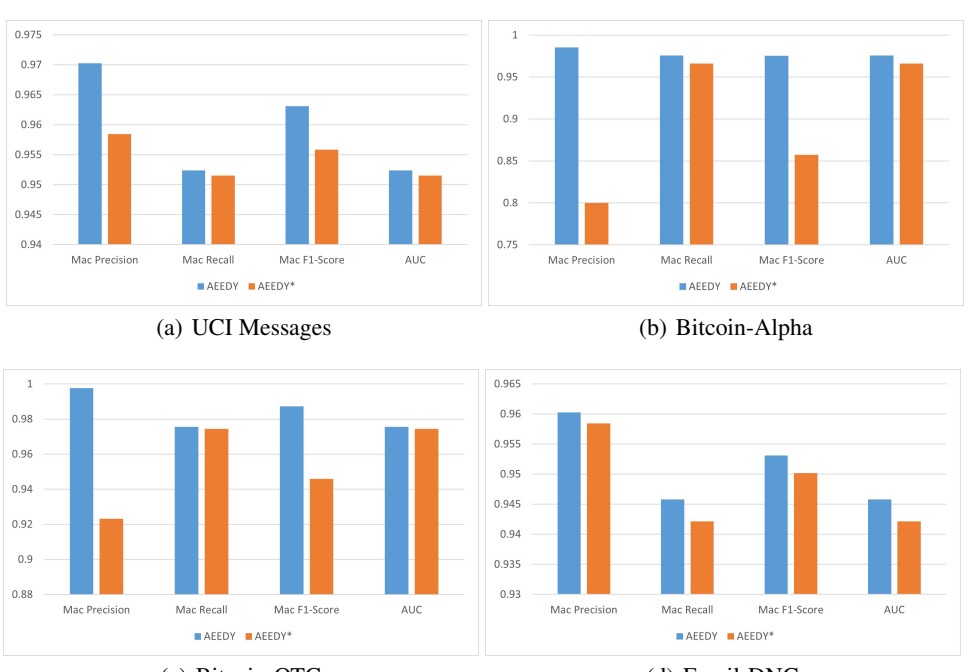

(a) UCI Messages        (b) Bitcoin-Alpha

(c) Bitcoin-OTC        (d) Email-DNC

Figure 5: Multi-indicator comparison results on the UCI Messages, Bitcoin-Alpha, and Email-DNC datasets.

To verify the effectiveness of adding discriminator $D$, comparative experiments are conducted on the UCI data, the Bitcoin-Alpha dataset, the Bitcoin-OTC dataset and Email-OTC with both discriminators removed and with discriminators, respectively. The four evaluation metrics, macro-average Precision, macro-average Recall and macro-average F1-score, AUC, are used for full identification. Using **AEEDY\*** to denote the model without discriminator $D$.The comparative results are shown in Figure 5. Based on the results of the experiment, it can be found that **AAEDY** has better indicators than **AEEDY\***.

## 5 CONCLUSION AND FUTURE WORK

Based on the idea of esidual analysis, this paper proposes a novel semi-supervised anomaly detection framework**AAEDY**, which solves the problem that the supervised model cannot be trained due to the scarcity of abnormal data. The framework mainly consists of two modules, the data preprocessing module and the anomaly addition detection module. The data preprocessing module uses three codes to obtain the spatiotemporal encoding information of the edge. The anomaly detection module consists of an adversarial code autoencoder, which performs anomaly detection by comparing the differences between edges in two low-dimensional spaces. The effectiveness of our proposed framework has been confirmed by experiments on six benchmark datasets. In the next step, we will combine node attribute information to jointly learn the complex cross-modal interactions between graph structure and node attributes.

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

---

**Algorithm 2** Model Test Algorithm

---

**Input:** Testing set of dynamic graph: $\mathbb{G}_{test} = \{\mathcal{G}^t\}_{t=1}^T$, Number of sampled contextual nodes: $k$,
    Size of time window: $\tau$, Number of attention heads $h$, Number of attention Layers $r$.
 1: Randomly initialize the parameters of encoding linear mappings, And use the already trained
    adversarial autoencoder model.
 2: Split the dynamic graph $\mathbb{G}_{test} = \{\mathcal{G}^t\}_{t=1}^T$ with maximum timestamp $T$ into snapshots at $\tau$
    timestamps $\mathcal{G}_{test}^t = (\mathcal{V}^t, \mathcal{E}^t) \in \{\mathcal{G}^t\}_{t=\tau}^T$
 3: **for** $e \in \mathcal{E}^t$ **do**
 4:    Set $e$ the as the target edge and sample its subgraph node set $\mathcal{S}(e)$ with $\tau(k+2)$
 5:    Calculate edge encoding matrix $X(e)$ via Equation (3)-(7)
 6:    Calculate the anomaly score with Equation (12)
 7: **end for**

---

# A   APPENDIX

## A.1   ALGORITHM

The overall testing process of our **AEDDY** framework is described in Algorithm 2. In Algorithm 2, step 1 is to initialize the parameters in the model, and in steps 2-7, step 2 is to divide the dynamic graph into several temporal snapshots, and step 4 is to perform preprocessing of the data to obtain the set of nodes of an edge. Step 5 is to obtain the fusion code of an edge by three encodings. In step 6, the anomaly value of the edge is calculated to determine whether it is an anomalous edge or not.

## A.2   DATASET DETAILS

Six real-world benchmark datasets of dynamic graphs are demonstrated as follows:

The UCI Messagegs dataset Opsahl & Panzarasa (2009) is a social network dataset consisting of users and the communication between them. In this dataset, there are 1,899 nodes, 13,838 edges, and the average degree of the nodes is 14.57.

The Bitcoin-Alpha dataset Kumar et al. (2016) and the Bitcoin-OTC dataset Kumar et al. (2018) are Bitcoin transaction data on the Bitcoin-Alpha and Bitcoin-OTC platforms, respectively, consisting of user and transaction information in the platforms, where the Bitcoin-Alpha dataset has 3,777 nodes and 24,173 edges, and the average degree of the nodes is 12.80. The Bitcoin-OTC dataset has 5,881 nodes and 35,588 edges, with an average degree of nodes of 12.10.

The Digg dataset De Choudhury et al. (2009) is a collection of social network information collected from August 3rd 2008 to August 6th 2008, consisting of information about users and inter-user communication on the Digg social networking site.

The AS-Topology Zhang et al. (2005) dataset is the network connections between autonomous systems of the Internet.

The Email-DNC dataset Rossi & Ahmed (2015) is an email network collected in 2016, which has 1,866 nodes with 39,364 edges in this dataset, and the average degree of the nodes is 42.08.

## A.3   BASELINES DETAILS

Seven anomaly detection methods are demonstrated as follows:

DeepWalk Perozzi et al. (2014) is inspired by Word2vec, which acquires node sequences by random wandering and performs feature embedding using skip-gram models.

node2vec Grover & Leskovec (2016) is a graph embedding method that combines DFS neighbourhoods and BFS neighbourhoods. In simple terms, it can be seen as an extension of deepwalk, a deepwalk that combines DFS and BFS random wandering.

Spectral Clustering Opsahl & Panzarasa (2009) is a technique for learning node embedding by maximizing the similarity between nodes in a region. The idea behind this strategy is to keep the local connection relationship between nodes in a network intact.

NetWalk Yu et al. (2018) obtains a sequence of nodes by designing a wander in a dynamic graph, embeds the nodes into a low-dimensional space using a deep self-encoder, and finally uses a technique of clustering to detect anomalous information.

AddGraph Zheng et al. (2019) based on the idea of residual analysis, trains a normal pattern by designing a semi-supervised learning framework that uses normal edges continuously during the training process. In this framework, GCN embedding features are mainly used to extract long and short-term spatio-temporal information using GRU.

StrGNN Cai et al. (2021) is also a deep learning-based framework for dynamic graph anomaly detection, mainly by using GCN to extract features from subgraphs under each temporal snapshot, and finally using the recurrent neural network GRU to fuse features from subgraphs under multiple temporal snapshots in order to capture the spatio-temporal information of the dynamic graph.

TADDY Liu et al. (2021) is a newly proposed anomaly detection framework for dynamic graphs. And uses graph diffusion techniques to sample sub-structures in dynamic graphs, and proposes three encoding methods to capture global structural information, local structural information and temporal information of the graph, and finally performs information fusion learning based on autoencoders to finally achieve anomaly detection on dynamic graphs.

