# OpenReview forum: "Anomaly Detection in Dynamic Graphs via Adversarial Autoencoder"
_ICLR.cc/2025/Conference — Submitted to ICLR 2025_

### Official Review · Reviewer_Q6Wy · 2024-10-21

**Soundness:** 2
**Presentation:** 1
**Contribution:** 2
**Rating:** 3
**Confidence:** 4

**Summary:**

This paper focuses on the problem of anomaly detection in dynamic graphs and proposes a semi-supervised method that only utilizes normal nodes during model training. The proposed method is based on an adversarial graph autoencoder. Empirical evaluations are conducted to demonstrate the effectiveness of the proposed method.

**Strengths:**

The idea of anomaly detection without labeled anomalous data is interesting and practical.

The proposed method achieves promising results over the existing baselines.

**Weaknesses:**

The introduction section needs to more clearly explain the connection between detecting abnormal edges and ensuring safety in financial transaction systems. Additionally, the contributions of the proposed method should be emphasized in a more focused way to highlight its significance.

The novelty of this paper is limited as it seems that it is a combination of TADDY and adversarial autoencoder.

The authors are encouraged to include a problem formulation section to clearly present the problem aimed to solve.

Some symbols are not explained clearly and timely such as the meaning of $C_{v_1}$ in Eq (2).

The related works and baselines are not sufficiently up-to-date. Most of them are published before 2021. The authors should include more recent research to ensure the paper reflects the current state of the field.

There are lots of grammar errors and typos. For example, the statements in line 55 and 56 are unclear. The authors should thoroughly proofread and refine the entire manuscript to improve its overall readability.

**Questions:**

Please see the weaknesses.

---

### Official Review · Reviewer_tm86 · 2024-11-02

**Soundness:** 3
**Presentation:** 3
**Contribution:** 2
**Rating:** 5
**Confidence:** 4

**Summary:**

For the detection of anomalous edges in dynamic graphs, a novel semi-supervised anomaly detection framework AAEDY based on adversarial idea is presented. First, the model obtains edge encoding of the target edge by the graph diffusion technique and three encoding methods, which aggregate the graph information at different times. Second, autoencoder reconstructs the target edges by global information, trains the model only by minimizing the errors between the original normal data and the reconstructed individual data, and tests anomalous edges by comparing the error between the original data and the reconstructed data.

**Strengths:**

AAEDY can detect anomalous edges by training the model without labelled anomaly data, which is often difficult to collect.
The edge encoding merges the spatial information and temporal information by three encoding methods.
The graph diffusion technique acquires connectivity top-k nodes of the target edge in time-sliding window, which combine the edge encoding with neighbor information at different times.
Adversarial loss, contextual loss and encoder loss are added together to form a single total loss function to ensure the model can reconstruct the edge, keeping away from fool the discriminator D.

**Weaknesses:**

On line 281, “resulting in a dissimilarity between z and zˆ” and “causing dissimilarity between z and zˆ”are repeated. Please revise this sentence to remove the repetition and express the idea more concisely.
Could you include a comparison to supervised method and provide empirical evidence or theoretical justification for whether the semi-supervised method is advanced without negative samples.
The autoencoder and the adversarial idea are not so novel. For example, autoencoder is used to anomaly detection in the artical “Variational Autoencoder based Anomaly Detection using Reconstruction Probability”, and only data with normal instances are used to train the autoencoder in this artical, which is similar to your approach. Could you explicitly discuss how your approach improves upon the VAE-based method mentioned above?

**Questions:**

How to set the proper threshold of anomaly score A(x test )? Is the threshold set by experiments or artificially? What happens to the AUC if the threshold is changed?
What is the time complexity of the process of Data Preprocessing?
Noting that it gives the best results with a number of subgraphs of 2 and 3 and a number of contextual nodes of around 7, is other previous moments and further neighborhood nodes redundancy?

---

### Official Review · Reviewer_1DVs · 2024-11-03

**Soundness:** 2
**Presentation:** 1
**Contribution:** 1
**Rating:** 3
**Confidence:** 5

**Summary:**

This paper presents a novel semi-supervised anomaly detection framework, AAEDY, designed to detect anomalous edges in dynamic graphs. The framework enhances reconstruction by incorporating adversarial techniques with an autoencoder and identifies anomalies by comparing original edges to reconstructed edges in a low-dimensional space.

**Strengths:**

S1. The paper is well-structured.

**Weaknesses:**

W1. In my opinion, the Introduction section does not clearly state the motivation behind this paper, nor does it adequately distinguish the AAEDY method from existing semi-/unsupervised dynamic graph anomaly detection approaches (beyond its suitability for large amounts of labeled anomaly data). Additionally, the challenges encountered in designing the AAEDY model are not discussed.
W2. The Related Work section lacks a comprehensive connection to current graph anomaly detection methods and omits unsupervised graph anomaly detection approaches based on contrastive learning.
W3. This graph anomaly detection method, based on diffusion, feels outdated and lacks originality. I do not believe it meets the standards of top-tier conferences like ICLR. The authors might consider submitting to a conference or journal of a slightly lower tier.
Other Issues
W4. There are several missing spaces on page two, for example, in the sentence: “In order to solve the problem that current dynamic graph anomaly detection methods are not very practical,AAEDY, a novel semi-supervisedadversarial proposes andynamic graph anomaly detection framework based onautoencoder, which only uses normal data to train the model.”
W5. Certain variables, such as top-k, need to be italicized, and each formula should be followed by appropriate punctuation.
W6. Is it AAEDY or AEEDY? The term "AEEDY" in Section 4.5 should be corrected to "AAEDY."

**Questions:**

Please refer to the weaknesses

---

### Official Review · Reviewer_TYas · 2024-11-03

**Soundness:** 2
**Presentation:** 2
**Contribution:** 2
**Rating:** 5
**Confidence:** 4

**Summary:**

The authors present a novel semi-supervised anomaly detection framework, AAEDY, designed for detecting anomalous edges in dynamic graphs. This framework enhances reconstruction by integrating adversarial techniques with autoencoders and discriminates between normal and anomalous edges by comparing the original edges to their reconstructed counterparts in a low-dimensional space.

**Strengths:**

1- The problem addressed is both interesting and relevant, tackling the critical issue of anomaly detection in dynamic graph structures.
2- The results demonstrate improved performance over baseline models, indicating that the proposed framework has potential effectiveness in real-world applications.

**Weaknesses:**

1- The paper lacks novelty, particularly in its reliance on autoencoders for unsupervised/semi-supervised anomaly detection, which is already prevalent in existing literature. Although the authors claim to propose three encoding methods, the intuition behind these methods is not clearly articulated. For instance, the importance of equation (3) in the context of anomaly detection is not sufficiently justified, nor is its superiority over existing encoding methods explained. Furthermore, the contributions outlined in the anomaly detection section (Section 3.2) are ambiguous, leaving readers uncertain about the framework's unique aspects. The authors’ claim regarding the addition of an embedding layer does not present significant novelty, and there is a lack of justification for why this approach was chosen or how it enhances the overall framework.
2-The methodology for generating abnormal data in the experiments is inadequately explained, with no clear introduction found in the main paper or appendix. Since the types of anomalies significantly affect results, the authors should consider employing multiple types of injection methods.
3-There are several writing issues throughout the paper, such as missing spaces between terms (e.g., "semi-supervised" and "adversarial" at line 055, "on" and "autoencoder" at line 055). These typographical errors detract from the overall professionalism of the manuscript and should be addressed in a thorough proofreading process.

**Questions:**

Could the authors provide more detailed information about the anomaly injection method, including the types of anomalies used and their potential effects on the results?

---

### Official Review · Reviewer_JEib · 2024-11-04

**Soundness:** 1
**Presentation:** 2
**Contribution:** 1
**Rating:** 3
**Confidence:** 5

**Summary:**

The paper presents AAEDY, a semi-supervised framework for anomaly detection in dynamic graphs, designed to mitigate the reliance on labeled anomaly data by training solely on normal instances. It combines an adversarial autoencoder with multi-level encoding techniques to capture spatial and temporal features, using residuals between original and reconstructed edges for anomaly detection. AAEDY demonstrates improved performance over baseline methods across six datasets in terms of AUROC on detecting injected anomalies.

**Strengths:**

The AAEDY framework demonstrates notable performance improvements on datasets with injected anomalies, suggesting its potential for anomaly detection in dynamic graphs. Its use of a “normal-only” training approach is also a relevant contribution, as it addresses the practical issue of limited labeled anomaly data in real-world settings.

**Weaknesses:**

Weaknesses:
1. Unclear technical contribution: The proposed framework includes multiple modules and, from my perspective, primarily combines known techniques without providing clear insights or justifications. For example, it’s unclear why both a discriminator enforcing the contextual loss to minimise the difference between the inputs and the reconstructions is needed.

2. Practicality of noise-free normal examples: How practical is it to ensure that normal examples are entirely free from noise? The default training set size is 80% and contains only normal data, which may not be very feasible. In general, annotating a handful of normal nodes is practical, but ensuring a large number of nodes are completely noise-free is challenging.

3. Scalability and runtime concerns: The proposed framework includes multiple modules, such as graph-based reconstruction and subgraph sampling, which are prone to limited scalability and long runtime. I believe it is important to discuss these issues in the paper.

4. Evaluation metrics: Although AUROC is used for evaluation, given the scarcity of the anomaly class, it is essential to include complementary metrics that focus on the anomaly class, such as AUPR, to better reflect performance for the anomaly class.

5. Hyperparameter settings: The hyperparameter settings vary significantly across different datasets. The author should clarify how these settings are determined and discuss their sensitivity.

6. Evaluation with realistic anomalies and dataset size: It appears that the anomalies used for evaluation are injected into datasets that lack actual anomalies, which may limit the realism of the results. Incorporating datasets with real-world anomalies would provide a more accurate assessment. Additionally, the datasets used are relatively small in terms of node count. It would be more insightful to evaluate the proposed methods on large-scale graphs.

7. Miscellaneous: (a). Please ensure consistent spacing. There are a few places where words are stuck together such as line 055 and 532. (b). The paper has multiple grammar errors, such as line 250 (Anomalous edge detection…) and line 275 (...does not be able to).
(c). I’m unable to find the architecture of the autoencoder in the method section. (d). Related works on static GAD only includes works until 2021. There are numerous static GAD works after 2021. (e) In general, the paper is not easy to follow, and the clarity should be further improved for ICLR's standard. (f) It would be better if the authors can provide more information on how the baseline methods are adapted to this normal-only GAD setting.

**Questions:**

Please refer to the weaknesses and address them accordingly if possible.

---

### Meta-Review · Area_Chair_MxZd · 2024-12-18

**Metareview:**

The work introduces a new method for semi-supervised anomaly detection in dynamic graphs. The method aims to improve the data reconstruction-based method for anomaly detection with an adversarial learning. The work receives five reviews, including three rejects (3) and two weak rejects (5). Overall, it has the following strengths:
- The studied setting — semi-supervised graph anomaly detection setting — is generally more practical than the widely studied unsupervised setting (JEib, TYas, tm86, Q6Wy)
- The method demonstrates effective performance on the datasets used (TYas, Q6Wy)

However, the reviewers find a number of major concerns:
- The proposed method lacks new technical insights and/or is not properly positioned and motivated (JEib, TYas, 1DVs, Q6Wy)
- The method requires a large number of noise-free normal samples during training, which can be difficult to obtain (JEib)
- The empirical evaluation is not convincing due to the lack of datasets with real anomalies and discussion on scalability,  and missing results on closely related baselines and popular evaluation metrics (JEib, TYas, tm86, Q6Wy)
- The method’s performance is sensitive to hyperparameter settings (JEib)
- The related work discussion misses important studies (1DVs, Q6Wy)

The reviews are consistent on the reject recommendation, and I generally agree with the concerns and the recommendation.

**Additional Comments On Reviewer Discussion:**

No rebuttal is submitted, and thus no author-reviewer discussion.

---

### Decision · Program_Chairs · 2025-01-22

Reject